# Genome-Wide Survey of the *RWP-RK* Gene Family in Cassava (*Manihot esculenta* Crantz) and Functional Analysis

**DOI:** 10.3390/ijms241612925

**Published:** 2023-08-18

**Authors:** Chenyu Lin, Xin Guo, Xiaohui Yu, Shuxia Li, Wenbin Li, Xiaoling Yu, Feng An, Pingjuan Zhao, Mengbin Ruan

**Affiliations:** 1School of Tropical Agriculture and Forestry, Hainan University, Haikou 570228, China; lcy1179456412@163.com (C.L.); guoxinyxh@hotmail.com (X.G.); xiaohuiyu@hainanu.edu.cn (X.Y.); 2Key Laboratory of Biology and Genetic Resources of Tropical Crops, Institute of Tropical Bioscience and Biotechnology, Chinese Academy of Tropical Agricultural Sciences, Haikou 571101, China; lishuxia@itbb.org.cn (S.L.); liwenbin@itbb.org.cn (W.L.); yuxiaoling@itbb.org.cn (X.Y.); 3Hainan Danzhou Agro-Ecosystem National Observation and Research Station, Rubber Research Institute of Chinese Academy of Tropical Agricultural Sciences, Danzhou 571737, China; an-f@catas.cn

**Keywords:** cassava, nodule inception-like proteins, RWP-RK domain proteins, abiotic stress, phylogenetic relationships, gene expression patterns, nitrogen starvation

## Abstract

The plant-specific RWP-RK transcription factor family plays a central role in the regulation of nitrogen response and gametophyte development. However, little information is available regarding the evolutionary relationships and characteristics of the RWP-RK family genes in cassava, an important tropical crop. Herein, 13 RWP-RK proteins identified in cassava were unevenly distributed across 9 of the 18 chromosomes (Chr), and these proteins were divided into two clusters based on their phylogenetic distance. The NLP subfamily contained seven cassava proteins including GAF, RWP-RK, and PB1 domains; the RKD subfamily contained six cassava proteins including the RWP-RK domain. Genes of the NLP subfamily had a longer sequence and more introns than the RKD subfamily. A large number of hormone- and stress-related cis-acting elements were found in the analysis of RWP-RK promoters. Real-time quantitative PCR revealed that all MeNLP1-7 and MeRKD1/3/5 genes responded to different abiotic stressors (water deficit, cold temperature, mannitol, polyethylene glycol, NaCl, and H_2_O_2_), hormonal treatments (abscisic acid and methyl jasmonate), and nitrogen starvation. MeNLP3/4/5/6/7 and MeRKD3/5, which can quickly and efficiently respond to different stresses, were found to be important candidate genes for further functional assays in cassava. The MeRKD5 and MeNLP6 proteins were localized to the cell nucleus in tobacco leaf. Five and one candidate proteins interacting with MeRKD5 and MeNLP6, respectively, were screened from the cassava nitrogen starvation library, including agamous-like mads-box protein AGL14, metallothionein 2, Zine finger FYVE domain containing protein, glyceraldehyde-3-phosphate dehydrogenase, E3 Ubiquitin-protein ligase HUWE1, and PPR repeat family protein. These results provided a solid basis to understand abiotic stress responses and signal transduction mediated by RWP-RK genes in cassava.

## 1. Introduction

Cassava (*Manihot esculenta* Crantz) is an important tropical and sub-tropical crop that provides food for over 800 million people worldwide because of its high starch production [1]. Cassava has a strong tolerance to drought and low-fertility soil environments because of its effective use of light, heat, and water resources [2]. However, the mechanism by which this crop displays high productivity under poor soil and abiotic stress conditions remains elusive. The *RWP-RK* transcription factor plays a central role in the absorption of nitrogen, which is an essential macronutrient for plant growth and development [3,4], but reports on the *RWP-RK* family in cassava are lacking. Thus, a systematic study of the *RWP-RK* gene family in cassava may help to elucidate cassava’s response to infertility and abiotic stresses.

The most prominent feature of the RWP-RK protein is a 60-amino-acid sequence of the RWP-RK domain that is conserved in all vascular plants, green algae, and slime molds [4]. The minus dominance (MID) protein is the first identified RWP-RK that is responsible for switching on and off the minus-plus programs in gamete differentiation [5]. Interestingly, the *MID* gene is also induced after N removal and activates gametogenesis [6]. Subsequently, the first nodule inception (NIN) protein is identified in the legume species *Lotus japonicus* as a crucial regulator that is required for the formation of infection threads and the initiation of primordia, and the inception of legume nodules is also dependent on the perception of N limitation in the plant [7]. Phenotypic analysis of pea (*Pisum sativum*) *sym35* (homologous gene of *PsNin*) mutants and *L.japonicus nin* mutants suggest that the *PsSym35* and *LjNin* genes have a similar function in early stages of root nodule formation [8]. To date, *RWP-RK* genes are widespread and preliminarily studied in algae, legumes, and non-legume plants [9,10,11].

RWP-RK proteins from different species have been classified into two subfamilies of NIN-like proteins (NLPs) and RWP-RK domain proteins (RKDs) depending on sequence [4]. *NLPs* mainly regulate the tissue-specific expression of genes involved in nitrogen, whereas *RKDs* mainly regulate the expression of genes involved in gametogenesis/embryogenesis [4,12,13]. In addition, NLPs have three typical characteristic domains, namely, RWP-RK, Phox and Bem1 (PB1), and GAF-like domains [4,13,14]. The PB1 domain is a ubiquitous protein–protein interaction domain, and it comprises another regulatory layer for NLPs via the protein interactions within NLPs or with other essential components [4,15]. GAF-like domains are detected at the N-terminal region of the NLP subfamily and might be involved in signal transduction [15,16]. Moreover, the RKD family is found in the genomes of green algae, vascular plants, and amoebozoa [4,17]. The RKD proteins contain a conserved RWP-RK domain and some other optional motifs. Furthermore, RKD proteins can be divided into three groups based on their optional motif differences [4]. In general, RKD proteins are remarkably shorter than NLP proteins from the same species because the PB1 and GAF-like domains may be absent [4,18].

The function mechanism of RWP-RK proteins has been extensively studied in different species including green algae, legumes, and vascular plants. NLP directly targets *NF-Y* subunit genes to control N-mediated symbiotic root nodule formation in *L. japonicus* [19]. *Arabidopsis thaliana* is the model plant used to study RWP-RK proteins [13,18]. *AtNLP7* plays an important role in early response to nitrate [19,20]. In addition, overexpression of *AtNLP7* enhances nitrogen and carbon assimilation and promotes plant growth under both restricted and sufficient nitrogen conditions [10]. *AtNLP8* is important to promote seed germination [21]. The nitrate-CPK-NLP signaling pathway and Ca^2+_^Ca^2+^ sensor protein kinase-NLP signaling cascades have been identified, and they allow NLPs to have central roles in mediating the nitrate signaling pathway [16,22]. Nitrate leads to the accumulation of NLP1 in the nucleus where it can suppress the ability of NIN to activate *CRE1* expression in *Nicotiana benthamiana* and *Medicago truncatula* [23]. RKDs controls germ cell differentiation during female gametophyte development in *Arabidopsis*, *Triticum aestivum*, and *Nicotiana tabacum* [11,18,24,25]. In summary, previous studies have highlighted the emerging roles played by NLPs in the N starvation response, N and P interactions [26,27,28], nodule formation in legumes [29,30,31,32], and root cap release in higher plants [32].

The increasing number of plant genomes sequenced has facilitated the study of the RWP-RK family in dicots and monocots, including *Zea mays* [33], *Brassica napus* [34], and *Malus domestica* [35]; however, information on RWP-RK in cassava has not yet been reported. In this study, cassava *RWP-RK* transcription factors were identified from the whole genome, and a phylogenetic tree based on the genome-wide gene duplication events was constructed. Gene structures, protein motifs, and the expression levels of *RWP-RK* genes in different cassava tissues and leaves of different stresses were systematically studied in detail. Subcellular localization and interaction of two RWP-RK proteins were investigated. The present results will provide important information for further study of the regulation mechanism of cassava RWP-RK genes under abiotic stresses.

## 2. Results

### 2.1. Cassava RWP-RK Protein Family

A total of 21 candidate proteins corresponding to the RWP-RK domain were originally obtained by searching query sequences. However, candidate proteins that contained partial RWP-RK domains were excluded. The cassava genome encoded 13 proteins carrying an RWP-RK domain, and these proteins were unevenly distributed across 9 of the 18 cassava Chr, with three members on Chr 2, two members on Chr 1 and 11, and one on each on Chr 3, 5, 6, 7, 14, and 18 (Figure 1). These proteins were divided into two classes: NLPs with similarity to NIN along their entire length and RKDs sharing only the RWP-RK domain with NIN. Seven proteins in the former class and six smaller proteins in the latter were named MeNLP1-7 and MeRKD1-6, respectively, based on their position on different Chr. Among the 13 RWP-RK proteins, MeRKD1 was the smallest protein with 120 amino acids, whereas MeNLP2 was the largest one with 1002 amino acids. Eighteen splice variants were predicted in MeNLP5, seven were predicted in MeNLP6, three were predicted in MeNLP4, and two were predicted in MeRKD3/5 and MeNLP7; the remaining six genes each produced a single transcript. MeNLP5/6 with many alternative splices can create more proteins with a slightly different function to regulate cassava development and growth [36].

### 2.2. Phylogenetic Analysis of the RWP-RK Family

We built a phylogenetic tree of RWP-RK proteins in *M. esculenta*, *O. sativa*, and *A. thaliana* to study the evolutionary relationships among the newly discovered RWP-RK proteins (Figure 2). Phylogenetic analysis indicated that RWP-RK proteins could be divided into four large groups. Among the four groups, group I contained the greatest number of RWP-RK proteins, and it was named the NLP subfamily because it included seven cassava proteins (MeNLP1-7), nine *Arabidopsis* proteins (AtNLP1-9), and five rice proteins (OsNLP1-5). In addition, group II was named the RKD subfamily because they included six cassava proteins (MeRKD1-6), six *Arabidopsis* proteins (AtRKD1-6), and two rice proteins (OsRKD2/4). Meanwhile, only OsRKD1 and OsRKD3 existed separately in groups III and IV, respectively, thereby indicating that the *RWP-RK* family genes had species-specific expansion in rice.

### 2.3. Synteny Analyses of RWP-RK Genes

Tandem and segmental duplications are considered the primary driving forces that expand gene families during evolution [37]. Herein, six *RWP-RK* genes were clustered into five duplication event regions on cassava linkage groups 1, 2, 6, and 14 (*MeNLP1/2*, *MeRKD1/3*, *MeRKD1/5*, *MeRKD1/6*, and *MeRKD5/6*; Figure 3). These results indicated that some *RWP-RK* genes were possibly generated by gene duplication. A chromosomal region within 200 kb containing two or more homologous genes is defined as a tandem duplication event [38,39]. The findings support that cassava has a highly repetitive genome that contains gene pairs with a large number of segmental duplications. Segmental duplications are the main source of adding new members to the cassava *RWP-RK* gene family.

A set of comparative syntenic graphs of cassava connected with dicots (*Arabidopsis*) and monocots (rice) was used to further infer the possible evolutionary implications of the cassava *RWP-RK* family. About 8 among the 13 identified *RWP-RK* genes showed no collinearity with either *Arabidopsis* or rice, suggesting that these homologous genes may have formed after the divergence of dicots and monocots or after the divergence of the two species. Nine and two cassava orthologous gene pairs were found in *Arabidopsis* and rice, respectively (Figure 4 and Appendix A). The two collinear genes of *MeRKD2* and *MeNLP7* were found between cassava and two other species (rice and *Arabidopsis*, *MeRKD2/OsRKD4/AtRKD5*, *MeNLP7/OsNLP2/MeNLP4*, or *MeNLP5*), which indicate that these collinear genes may already have existed before the ancestors of monocot and dicot separation. More syntenic gene pairs were found between cassava and dicots than between cassava and monocots. For example, the collinear gene pairs of *MeNLP2/AtNLP6*, *MeRKD6/AtRKD1/2*, *MeRKD2/AtRKD5*, and *MeRKD5/AtRKD2/1* were not found between cassava and rice, which may indicate that these orthologous pairs formed after the divergence of dicotyledonous and monocotyledonous plants.

### 2.4. RWP-RK Protein Structure and Conserved Motif Analysis

Thirteen proteins were divided into three categories by the MEME/MAST program to better understand the structural variability of RWP-RK proteins (Figure 5A). The amino acid sequences of 10 different motifs are shown in Figure 5B and Appendix A. Sequence alignment results showed that motifs #2/#8/#10 were different conserved RWP-RK domains that played a key role in the N-mediated control of development. Motifs #4/#7 were conserved PB1 domains, comprising approximately 80 amino acid residues. Sequence alignment showed that motifs #1/#3/#5/#7 were conserved GAF domains. Seven proteins that belonged to the first category contained 9 of all 10 motifs (except for MeNLP5 which included 7 motifs). The second category only contained one protein of MeRKD4 which only contained one RWP-RK motif. The last category included five proteins of MeRKD1/2/3/5/6 which contained two RWP-RK motifs, except for MeRKD5, which included only one motif that was absent in MeNLP1-7 and MeRKD4 proteins. RWP-RK proteins in the same category shared similar motifs, thereby suggesting that similar RWP-RK members have conserved biological functions [34]. We aligned the coding sequences to analyze the exon/intron structures of *RWP-RK* genes (Figure 5C). Genes belonging to the first category had complete sequences with 4–6 exons and 3–5 introns, respectively, including UTR regions at both ends (except for *MeNLP5*). Gene sequences in the remaining category were incomplete with 2–5 exons and 1–4 introns, respectively, without a UTR region (except for *MeRKD1*). Therefore, the average number of exons was greater in the *NLPs* (average number of 4.7) than in the RKDs (average number of 3.3). The similar gene structures observed in the same *RWP-RK* subfamily were consistent with their phylogenetic relationships.

### 2.5. Analysis of Cassava RWP-RK Gene Promoters

Cis-acting regulatory elements are important molecular switches involved in the transcriptional regulation of a dynamic network of gene activities controlling various biological processes, including hormone response, abiotic stress response, and development [40]. A large number of hormone-related cis-acting elements were identified through the analysis of cassava *RWP-RK* promoters (Figure 6), including ABA-responsive element (ABRE) in response to ABA [40,41], homeodomain-leucine zipper (HD-Zip) in response to auxin [42], and activation sequence-1 (as-1) in response to SA [43]. Furthermore, cis-acting elements related to adversity (e.g., low temperature, drought, and salt stress) were found, such as MYB, MYC, and MYC-like [44,45]. In addition, the TCT motif was required for the transcription of ribosomal protein gene promoters [46]. MYB and MYC recognition sites were present in almost all the genes analyzed (Figure 6), and MYB sites in a dehydration-responsive gene (RD22) promoter were proven to be cis-acting elements in the drought-inducible expression of RD22 in transgenic tobacco [47]. MYC sequence CATGTG played an important role in the dehydration-inducible expression of the *Arabidopsis EARLY RESPONSIVE TO DEHYDRATION STRESS* gene [45].

### 2.6. Transcription Profiles of RWP-RK Genes in Different Cassava Tissues

Specific primers were designed based on the conserved sequence to analyze the transcription activity of 13 *RWP-RK* genes in different tissues (Figure 7). *MeRKD2/6* genes were not expressed in all the tissues tested. The expression level of *MeRKD3/5* genes were lower in stems than in leaves, and the rest were close to leaves. Seeds and storage roots had more genes with low expressions levels than leaves, such as *MeRKD1/3/4/5* and *MeNLP2/3/6/7/* genes in seeds, and *MeRKD1/3* and *MeNLP1/2/3/5/7* genes in storage roots. Only *MeNLP1/5* genes were high in seeds, *MeRKD4/5* and *MeNLP6* genes were high in storage roots, and only *MeNLP4* in seed was close to the leaves. The expression levels of eight genes in flowers and fruits were similar; for example, *MeNLP3/6/7* and *MeRKD1/3/4* genes were low, *MeNLP1* and *MeRKD5* genes were high in two tissues, and only the *MeNLP2* gene was high in flower and low in fruit. Most genes with high expressions were found in roots, such as *MeNLP1/3/4/5/7* and *MeRKD1*/5. *MeNLP3/7* genes, which were an evolutionarily conserved gene pair, had a similar expression pattern in different tissues, but the expression levels of *MeNLP1/2* genes (an evolutionarily conserved gene pair) were completely different.

### 2.7. Expression Patterns of RWP-RK Genes in Response to Different Stresses in Cassava

We hypothesized that *RWP-RK* genes may respond to stress and hormone treatments because their promoter regions contain hormone- and stress-related motifs. RNA was isolated from leaves treated with drought, cold, salt, osmotic, and H_2_O_2_ stress to identify the abiotic-responsive *RWP-RK* genes in the cassava genome. In addition, the relative expression patterns of 13 cassava *RWP-RK* genes were evaluated with qRT-PCR. Nevertheless, *MeRKD2/6* genes were not detected, and the *MeRKD4* gene showed no significant difference in all samples. The relative expression levels of 10 genes detected were significantly down- or up-regulated in one of the three treatment groups (DM, DL, and RW) compared with those in the control group. The gene expression levels of *MeNLP1/2/4/6* and *MeRKD1/3/5* were significantly higher than the control at some point during the drought period, the expression level of *MeNLP3/5* was significantly lower than that of the control, and *MeNLP7* and *MeRKD1* exhibited an increased and then depressed expression pattern (Figure 8A). *MeNLP3* and *MeRKD5* genes were up-regulated in one or all the processing points when cold treatment lasted from 0.5 to 12 h, whereas *MeNLP1/2/5/6/7* and *MeRKD3* were down-regulated in one or all the treatment groups. *MeNLP4* increased and then decreased, and *MeRKD1* showed the opposite trend. Of all the genes, *MeRKD5* gene expression increased the most significantly during the whole cold period (Figure 8B). Ten *RWP-RK* genes responded to NaCl treatment. *MeNLP1/3* genes were uniformly significantly down-regulated at all points when NaCl treatment lasted from 1 h to 12 h, *MeNLP4* and *MeRKD1/5* genes were significantly up-regulated at some point with the same treatment, and *MeNLP2/5/6/7* and *MeRKD3* genes exhibited a dynamic change (Figure 8C). Out of those ten genes, the responses of the nine *RWP-RK* genes to Man and PEG were highly similar. *MeNLP4/MeRKD1* and *MeNLP4* genes did not respond to PEG and Man during treatment, respectively, and the rest were uniformly significantly down-regulated in some time points when Man and PEG treatments lasted from 1 h to 12 h (Figure 8D,E).

We obtained the data of 11 *RWP-RK* genes for H_2_O_2_ treatment. *MeNLP3/7* and *MeRKD3* genes were uniformly significantly down-regulated in every time point, *MeNLP1/2/5/6* and *MeRKD1* genes were down-regulated at some point when H_2_O_2_ treatment lasted from 1 h to 12 h; only the *MeNLP4* gene exhibited dynamic changes, and the *MeRKD5* gene was significantly up-regulated at some point (Figure 8F).

### 2.8. Expression Patterns of RWP-RK Genes in Response to Different Hormone Treatments in Cassava

Herein, we obtained the data of 10 *RWP-RK* genes for ABA and MeJA treatments, but *MeRKD2/4/6* genes were not detected (Figure 9A,B). Our results showed that *MeNLP4* and *MeRKD1* genes were induced in some time points by ABA application, whereas *MeNLP1* and *MeRKD3/5* genes exhibited a depressed expression pattern at some time points. Meanwhile, *MeNLP2/3/5/6* genes demonstrated early up-regulation after 1 h, followed by reduced expression after 3 h. *MeNLP7* expression was lower after 1 and 3 h but promptly increased after 6–12 h (Figure 9A). Nine genes responded to MeJA treatment. *MeNLP1/3/6* genes were down-regulated at every point when MeJA treatment lasted for 1 h to 12 h. By contrast, *MeNLP4* and *MeRKD1* genes were up-regulated in every point, and *MeNLP2/7* genes were up-regulated only after 6 h and 3 h, respectively. *MeNLP5* was down-regulated after 1–3 h, and peaks occurred only after 6 h. *MeRKD3* was up-regulated after 3 h, and it continued to decline thereafter (Figure 9B).

### 2.9. Expression Patterns of RWP-RK Genes in Response to Nitrogen Starvation Treatments in Cassava

The response of *RWP-RK* family genes to nitrogen starvation was studied using the leaves and roots of three- and six-day transplanted seedlings in nitrogen starvation culture as samples. The levels of *MeNLP2/3/4/5/7* and *MRKD1* in the root were consistently up-regulated at all treatment samples. *MeNLP6* and *MeRKD3/5* were also up-regulated in three- and six-day samples, respectively; only *MeNLP1* expression was inhibited in three-day samples (Figure 10A,B). Meanwhile, the gene expression levels of *MeNLP1/2/3/5/6/7* and *MeRKD1/3* in leaves were up-regulated in six-day samples (Figure 10C). Notably, *MeNLP3/5/7* displayed prominent changes in expression levels in the root. Remarkably, *MeRKD3* and *MeRKD5* showed the most significant improvement in the root and leaf, respectively.

### 2.10. Subcellular Localization of MeNLP6 and MeRKD5

Tobacco leaf epidermal cells expressing GFP showed cytoplasmic, plasma membrane, and nuclear localization (Figure 11A). However, the GFP signals in MeRKD5 and MeNLP6 fusion-expressing cells could be clearly detected in the nucleus with nuclear localization signal (Figure 11B,C). These observations confirmed that both MeRKD5 and MeNLP6 were localized to the nucleus.

### 2.11. Proteins Interacting with NLP6/RKD5 by Yeast Two-Hybrid System

The interaction proteins of MeNLP6 and MeRKD5 were screened in a cassava nitrogen starvation library with a yeast two-hybrid system. The results showed that five and one candidate proteins, respectively, were screened interacting with MeRKD5 and MeNLP6 (Table 1). The five proteins were agamous-like mads-box protein (MeAGL14), metallothionein-2 protein (MeMT2), Zine finger FYVE domain containing protein (MeZF), glyceraldehyde-3-phosphate dehydrogenase (MeGAPD), and E3 Ubiquitin-protein ligase HUWE1 (MeUBA), and the one protein was PPR repeat family protein (MePPR).

## 3. Discussion

A total of 13, 15, and 9 proteins containing the RWP-RK domain were identified in *M. esculenta*, *A. thaliana*, and *O. sativa*, respectively. There was one more RWP-RK protein than the 14 found in previous reports in *Arabidopsis* [4,14]. AT5G16100, which was discovered to include the RWP-RK motif, was named AtRKD6 (Appendix A). Previous reports on *O. sativa* revealed 7 and 16 RWP-RK proteins [4,14]. We identified nine proteins in rice, and they were named by comparing them with previous reported protein sequences (Appendix A). OsNLP1/4/5 and OsRKD1/2/4 followed the same names that were previously reported [48]. OsNLP2/3 in 2005 was later identified as the same protein named as OsNLP2, and OsRKD3 was matched with OsRKD8/9 in 2015; the rest of the rice RWP-RK proteins in 2014 were repeated or not matched.

A phylogenetic tree of RWP-RK proteins in *M. esculenta*, *O. sativa*, and *A. thaliana* indicated that RWP-RK proteins could be divided into four large groups. OsRKD2 and OsRKD4 existed separately in groups III and IV, respectively. RWP-RK proteins from cassava and *Arabidopsis*, two dicotyledonous plants, could be divided into groups I and II (Figure 2). On the basis of our analysis, MeNLP1-7, AtNLP1-9, and OsNLP1-5 proteins in group I were named as the NLP subfamily because these proteins harbored RWP-RK, PB1, and GAF domains; these results were consistent with the typical molecular structure of NLPs [14,28]. Meanwhile, MeRKD1-6, AtRKD1-5, and OsRKD2/4 proteins in group II were named as the RKD subfamily because these proteins harbored the RWP-RK domain and did not show any PB1 domains, and the GAF domains were random [14,48]. The number of NLPs was higher than that of RKDs in non-nodulating plants, such as nine NLPs and six RKDs in *Arabidopsis*, five NLPs and four RKDs in rice, and seven NLPs and six RKDs in cassava (Figure 1 and Figure 5). The gene sequence of the *NLP* subfamily was longer and more complete with more introns as compared to that of the RKD subfamily. The average number of exons was greater in the *NLPs*, with 4.7 average exons (4–6), than in the *RKDs,* with only 3.3 average exons (2–5). The number of exons in 18 *TaNLP* genes was generally 4 or 5, and most of the 19 *TaRKD* genes did not differ much from 3 to 5 exons [11]. Similar gene structures observed in the same *RWP-RK* subfamily were consistent with their phylogenetic relationships.

Gene duplication helps expand the gene family [49]. Collinearity analysis indicated that *NLP1*, *RKD1*, *RKD3*, and *RKD5* had duplicate genes in cassava. Those segmental duplication events were important in the evolution of cassava *RWP-RK* genes (Figure 3). Syntenic maps of cassava associated with dicots (*Arabidopsis*) and monocots (rice) showed that two and five cassava *RWP-RK* genes, respectively, had a syntenic relationship with two rice and six *Arabidopsis* genes. Thus, some orthologous pairs formed after the divergence between dicotyledonous and monocotyledonous plants.

In general, *RKDs* are highly expressed in reproductive organs, thereby highlighting their regulatory importance in female gametophyte development in *Arabidopsis* [18]. In cassava, only the *MeRKD5* gene was highly expressed in flowers, whereas the other *RKDs* had low expression in flowers and fruits. However, *MeRKD2/6* genes were not detected in any cassava tissues, and all the genes detected were lowest in seeds. The expression levels of *MeRKD1* in root, *MeRKD3* in leaf, and *MeRKD4/5* in root tuber were higher than those in the other parts (Figure 7). The promoter region of *RKDs* contained ABRE or/and SYB or/and HD-Zip elements in response to hormones or/and stresses (Figure 6). These results suggested that *MeRKDs* may also play a role in plants in response to stress. *MeNLP1/3/4/5/7* and *MeNLP6* genes showed the highest expression in root and storage roots, respectively. Strangely, the expression levels of *MeNLP1/2/3/7* were higher in flowers than in other parts.

Several studies on *NLP* gene function focused on nitrogen absorption [23,35,50]. In *Arabidopsis*, the expression levels of most *AtNLP* genes were induced by low nitrogen/nitrate starvation in seedlings; for example, *AtNLP1/3* genes were induced by KNO_3_, and *AtNLP5/8/9* genes were repressed during the same treatment [14,51]. OsNLP4 can affect the nitrate reductase activity and nitrate-dependent growth of rice by regulating gene expression within the nitrate signaling pathway [48,52]. Most *ZmNLP* genes demonstrated regulation with decreased expressions levels at 1 and 2 h after nitrate treatment and increased expression levels at 0.5 and 1.5 h [53]. Some experiments on N stress were also conducted in the study. Almost all of the *NLP* genes and *RKD1/3/5* gene in cassava root or leaf were extremely significantly induced in nitrogen-free culture, which indicates that participation in nitrogen stress remains the main function of *MeNLP* genes. Surprisingly, *MeRKD1/3/5* genes that play an important role in gamete formation may have the same function as *MeNLP* [12,17].

*NLP* genes can also respond to various stresses; for example, *AtNLP4/8* expression levels were induced by abiotic and chemical stresses and hormonal treatments [14], and *OsNLP4* expression in rice is repressed by drought/cold/heat abiotic stresses and induced by low phosphate availability [14]. In the present study, we completed seven different treatments on cassava leaves, and *MeRKD2/4/6* genes were not detected or no significant changes in all treatments were observed. Furthermore, *MeNLP4* expression after PEG and Man treatments and *MeRKD5* expression after MeJA treatment were not significantly changed. All the other cassava *RWP-RK* genes were significantly induced/repressed by multiple treatments in the experiments. The different variation trends in the expression levels of 10 genes at different periods of the same treatment indicated that the different *RWP-RK* genes had various responses to the same treatment. Interestingly, the transcript levels of *MeRKD1/3/5* genes were significantly different between the control and nine treatment groups. The results showed that these genes were associated not only with powdery spores but also with stresses.

As previously reported, MeRKD5 and MeNLP6 proteins were localized to the cell nucleus in tobacco leaf [12,21].

Five proteins interacting with MeRKD5, MeAGL14, MeMT2, and MeZF may interact with MeRKD5 alone or together to modulate the expression of N metabolism genes by binding the NRE in their promoters [54,55,56,57,58]. MeGAPD was down-regulated while MeUBA was up-regulated in cassava leaves under drought stress [59]. This result indicates that MeGAPD, MeUBA and MeRKD5 proteins may be involved in response to drought in cassava together. UBA was also found to interact with GAPD in wheat [60]. MePPR interacting with MeNLP6 had not been identified in previous RWP-RK studies.

## 4. Materials and Methods

### 4.1. Identification of RWP-RK Family Members in Cassava

Herein, 15 *Arabidopsis* RWP-RK protein sequences were downloaded from NCBI by searching RWP-RK and previous reports [4,14]. The cassava protein database was obtained from JGI (http://phytozome.jgi.doe.gov (accessed on 30 May 2022), *Manihot esculenta* V8.1), and rice (*Oryza sativa*) RWP-RK proteins were obtained by searching from EnsemblPlants (http://plants.ensembl.org/RWP-RK (accessed on 1 June 2022)) and previous reports [4,14,48]. Two different approaches were used to identify the cassava RWP-RK family proteins as follows: firstly, fifteen *Arabidopsis* RWP-RK amino acid sequences and all cassava protein sequences were used as query sequences and a library for query sequences, respectively, and a local BLASTP search with E-value ≦ 1 × 10^−5^ was performed; secondly, the query strategy based on HMM was derived from the PFAM database (http://pfam.xfam.org/ (accessed on 2 June 2022)) to download the protein conserved domain RWP-RK file (PF02042) and execute the HMM search command E-value ≦ 1 × 10^−5^. Cassava RWP-RK family members were finally obtained by confirming the RWP-RK domain with Batch CD-Search in NCBI (https://www.ncbi.nlm.nih.gov/Structure/cdd/cdd.shtml (accessed on 3 June 2022)) and SMART (https://smart.embl.de/ (accessed on 3 June 2022)). The location data of cassava RWP-RK proteins were also obtained from the genome information. MapChart2.32 (https://www.wur.nl/en/show/mapchart.htm (accessed on 5 June 2022)) was used to construct chr distribution map.

### 4.2. Phylogenetic Analysis of RWP-RK Family Members

The RWP-RK families containing the RWP-RK domain were identified to have 13, 15, and 9 proteins in *M. esculenta*, *A. thaliana*, and *O. sativa*, respectively (Appendix A). Cassava RWP-RK proteins were named based on their positions on Chrs and their included motifs (e.g., MeNLP1-7 and MeRKD1-6). RWP-RK proteins of *Arabidopsis* and rice were named by protein annotation (AtNLP1-9, AtRKD1-6, OsNLP1-5, and OsRKD1-4). Phylogenetic analysis was conducted using MEGA7 software (MEGA 7.0.21). The neighbor-joining method was used with 1000 bootstrap replications.

### 4.3. Cassava RWP-RK Gene Duplication

Gene duplication is ubiquitous and plays an important role in plant evolution. A gene duplication landscape was obtained using MCScanX (http://chibba.pgml.uga.edu/mcscan2/ (accessed on 15 July 2022)). Each duplicate segment with *RWP-RK* family genes was selected, and the syntenic map was generated using CIRCOS (http://circos.ca/software/download (accessed on 15 July 2022)).

### 4.4. Cassava RWP-RK Gene Structure and RWP-RK Protein Motif Analysis

The gene structure data used in this study were obtained from the genome annotation file (http://phytozome.jgi.doe.gov (accessed on 20 July 2022), *Manihot esculenta* V8.1) and the *Arabidopsis* Information Resource website (TAIR, http://www.arabidopsis.org/browse/genefamily/RWP-RK.jsp (accessed on 20 July 2022)). Gene structure analysis was conducted using Gene Structure Display Server version 2.0 (http://gsds.cbi.pku.edu.cn/ (accessed on 20 July 2022)). The motifs of RWP-RK proteins were analyzed using the MEME program (http://meme-suite.org/tools/meme (accessed on 24 July 2022)) based on the amino acid sequences. The motif distribution type was zero or one occurrence per sequence, and only motifs with E-values > 0.05 were present.

### 4.5. Cassava RWP-RK Gene Promoter Analysis

A 1500 bp sequence of the *RWP-RK* gene translation initiation site upstream was extracted from the cassava genome as the promoter region. All cassava *RWP-RK* promoter sequences were submitted to plantCARE database (http://bioinformatics.psb.ugent.be/webtools/PlantCare/HTML/ (accessed on 9 May 2023)) to predict the cis-acting elements of the promoter.

### 4.6. Transcription Profile of Cassava RWP-RK Genes in Different Tissues

Tissues of leaves, stems, roots, flowers, storage roots, and fruits were collected from field-grown South China 124 (SC124) plants (360 days old) for further RNA isolation to study the expression profiles of cassava *RWP-RK* genes in different tissues. SC124 was planted in Hainan China (20°02′ N, 110°35′ E).

### 4.7. Drought and Cold Treatment

To analyze the expression levels of *RWP-RK* genes in response to drought stress, SC124 plants were grown in a plastic pot (35 cm diameter × 25 cm height), and the substrate was made by mixing two parts latosolic red soil with sand (mixing 1:1). The plants were grown in a non-air-conditioned glass house with a temperature between 19 and 33 °C and sunlight, and the plants were watered two to three times every week and fertilized with a fertilizer (N:P:K = 15:15:15). Eighty-day-old plants were treated by withholding water for 14 and 18 days as drought in middle (DM) and drought in later (DL), respectively. Some plants were re-watered 24 h after 18 days of drought treatment as a re-watering sample (RW). Plants watered as normal were used as controls. Leaves were collected from the three plants. Three leaves (the second, third, and fourth leaves from the apex) were collected and mixed from each plant as a sample.

To analyze the *RWP-RK* gene expression levels in response to cold stress, 60 days old SC124 potted plants were kept at 4 °C (cold treatment) and 25 °C (control) with a 16 and 8 h light/dark cycle and 100 μmol m^−2^ s^−1^ light intensity to study the expression profile of *RWP-RK* genes in response to cold stress. Treated leaves were collected as samples at 0.5, 1, 3, and 12 h after cold treatment with three biological replicates.

### 4.8. Different Salt, Osmotic, Oxidation and Hormonal Treatments

Mature leaves from SC124 potted plants (60 days old) were placed on sterile filter paper with 7% polyethylene glycol 6000 (PEG), 300 mmol mannitol (Man), 100 μmol abscisic acid (ABA), 100 μmol methyl jasmonate (MeJA), 10 mmol H_2_O_2_, and 10 mmol NaCl solutions, and kept in a Petri dish at room temperature for 24 h to study the expression profile of *RWP-RK* genes in response to different treatments. The control leaves were placed on sterile filter paper with water. The samples were collected at 1, 3, 6, and 12 h after treatment with three biological replicates.

### 4.9. Nitrogen Starvation Treatments

One-month-old cv.60444 cassava seedlings cultured on MS medium were transplanted to MS medium in nitrogen-free conditions. Control seedlings were transplanted to MS medium. The root samples were collected for 3 and 6 days, and the leaf samples were collected for 6 days after treatment with three biological replicates.

### 4.10. RNA Extraction and qRT-PCR Analysis

All samples were collected, immediately preserved in liquid nitrogen, and stored at −80 °C for RNA extraction. Total RNA was extracted using the RNAprep Pure Plant Kit (DP432, Tiangen Biotech, Beijing, China) following the manufacturer’s instructions. The cDNA used to verify exon/intron structure was synthesized using HiScript^®^ II 1st Strand cDNA Synthesis Kit (R211-01, Vazyme Biotech Co., Nanjing, China). Herein, to detect transcription levels of the *RWP-RK* genes in different samples, specific primers for each gene were designed through NCBI Primer-BLAST (http://www.ncbi.nlm.nih.gov/tools/primer-blast/ (accessed on 20 July 2022), Appendix A), and PCR was conducted in triplicate for each cDNA sample by using an SYBR^®^ Premix Ex Taq™ II Kit FP205-02 (TIANGEN Biotech Co., Ltd., Beijing, China) on a StepOne™ Real-Time PCR system (Thermo Fisher scientific, Waltham, MA, USA) with the following thermal cycling profiles: 95 °C for 10 min, 40 cycles of 95 °C for 10 s, and 60 °C for 30 s. The relative expression levels of the *RWP-RK* genes were calculated using the 2^−ΔΔCt^ method. Statistical difference was determined via one-way ANOVA.

### 4.11. Subcellular Localization of MeNLP6 and MeRKD5

To investigate the subcellular localization of MeNLP6 and MeRKD5, the 35S:MeNLP6-GFP and 35S:MeRKD5-GFP fusion constructs were produced by inserting the full length CDS of MeNLP6 and MeRKD5 into the pCambia1302-35S: GFP vector, and the resulting vectors were transformed into *Agrobacterium* LBA4404, transiently introduced into tobacco leaf epidermal cells via *Agrobacterium* injection, and imaged under a confocal laser scanning microscope (Olympus FluoView FV1100).

### 4.12. Yeast One-Hybrid Assay (Y2H)

The cDNA of the nitrogen-free samples was obtained, and the cDNA and pGADT7 (AD)-Rec were co-transformed into yeast strain Y2HGold to directly construct the library and prepare competent cells. Two cDNA fragments encoding MeNLP6 and MeRKD5, respectively, were amplified and inserted into plasmid pGBKT7 Vector (BD) to get BD/MeNLP6 and BD/MeRKD5. The constructed vectors were co-transferred into yeast Y2HGold competent cells, and the cells were grown on SD-Trp-Leu medium for 3 days at 30 °C, then transferred to SD-Trp-Leu-His medium with 200 μM AbA^+^ at different dilutions. The yeasts were incubated at 30 °C for 5 days and the extent of yeast growth was determined. The AD and BD empty vector were used for negative control.

## 5. Conclusions

Genome-wide analysis of cassava revealed 13 *RWP-RK* family genes, which could be divided into *NLP* and *RKD* subfamilies including seven and six genes, respectively. The RWP-RK subfamily members had similar gene structures and motif compositions, and the *NLP* subfamily genes had more introns and motifs than the other subfamily. The evolution of cassava *RWP-RK* genes was affected by segmental duplication events. Six *MeNLP* genes showed higher expression levels in the root (including storage root) than in the leaf, but *RKD1/4/5* genes had high expressions levels in non-reproductive organs. A large number of hormone- and stress-related cis-acting elements, which were activated by a range of stimuli, were found through the analysis of *RWP-RK* promoters. Expression profile analyses revealed that almost all *MeNLP* and *MeRKD1/3/5* genes were susceptible to six abiotic stressors (drought, cold, salt, PEG, Man, and H_2_O_2_) and hormonal (ABA and MeJA) treatments. MeNLP3/5/7 and MeRKD3/5 genes, which displayed the most prominent changes in expression levels under nitrogen starvation, were the candidate genes to create an excellent new germplasm. The MeRKD5 and MeNLP6 proteins were found to localize to the cell nucleus in tobacco leaf. Five and one candidate proteins interacting with MeRKD5 and MeNLP6 were screened from a cassava nitrogen starvation library; these findings may be beneficial for studying RWP-RK gene function in cassava.

## Figures and Tables

**Figure 1 ijms-24-12925-f001:**
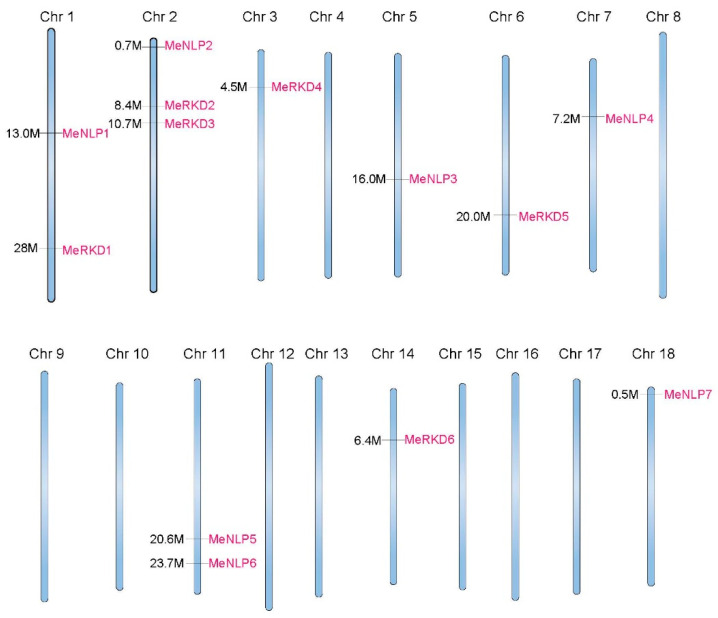
Location of 13 cassava RWP-RK proteins in chromosome.

**Figure 2 ijms-24-12925-f002:**
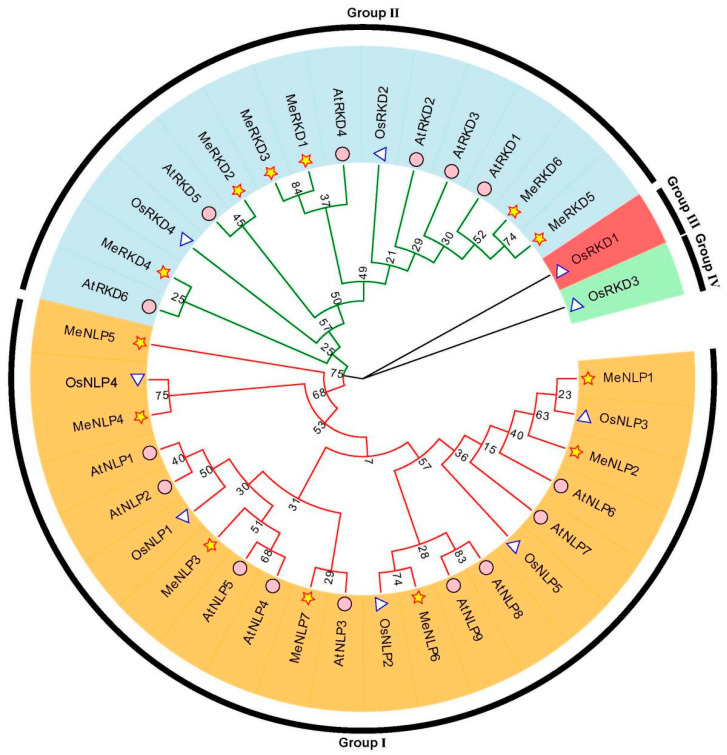
Phylogenetic evolution of RWP-RK proteins in *M. esculenta*, *A. thaliana* and *O. sativa*. The yellow pentagram represents cassava RWP-RK proteins, the white triangle represents rice RWP-RK proteins, and the purple circle represents *Arabidopsis* RWP-RK proteins. The numbers at nodes represent bootstrap values per 1000 replicate determined with the neighbor-joining (N-J) method.

**Figure 3 ijms-24-12925-f003:**
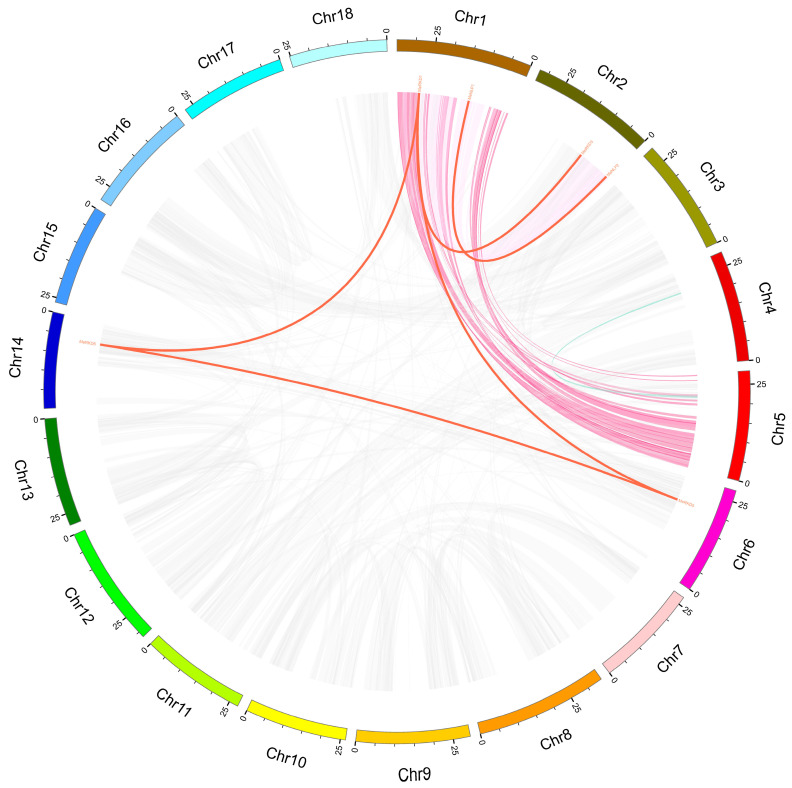
Schematic representations for the chromosomal distribution and inter chromosomal relationships of cassava *RWP-RK* genes. Gray lines indicate all synteny blocks in the cassava genome, and the red lines indicate duplicated RWP-RK gene pairs. The chromosome number is indicated at the bottom of each chromosome.

**Figure 4 ijms-24-12925-f004:**
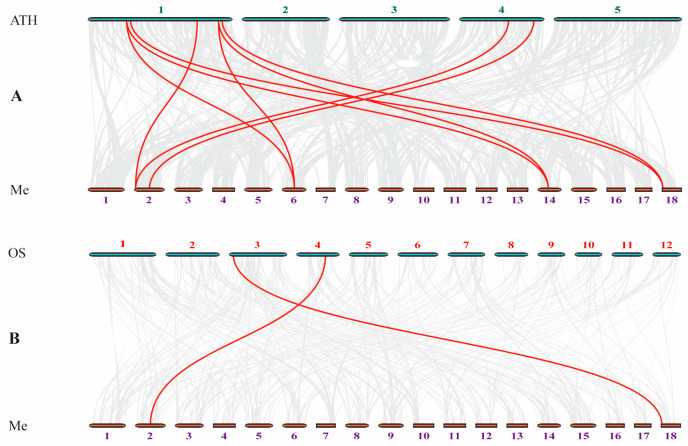
Synteny analysis of RWP-RK genes between cassava and two representative plant species. Gray lines in the background indicate the collinear blocks within cassava and other plant genomes, while the red lines highlight the syntenic cassava gene pairs with *A. thaliana* and *O. sativa*, respectively. The purple, red and green number represent cassava, rice and *Arabidopsis* chromosome, respectively.

**Figure 5 ijms-24-12925-f005:**
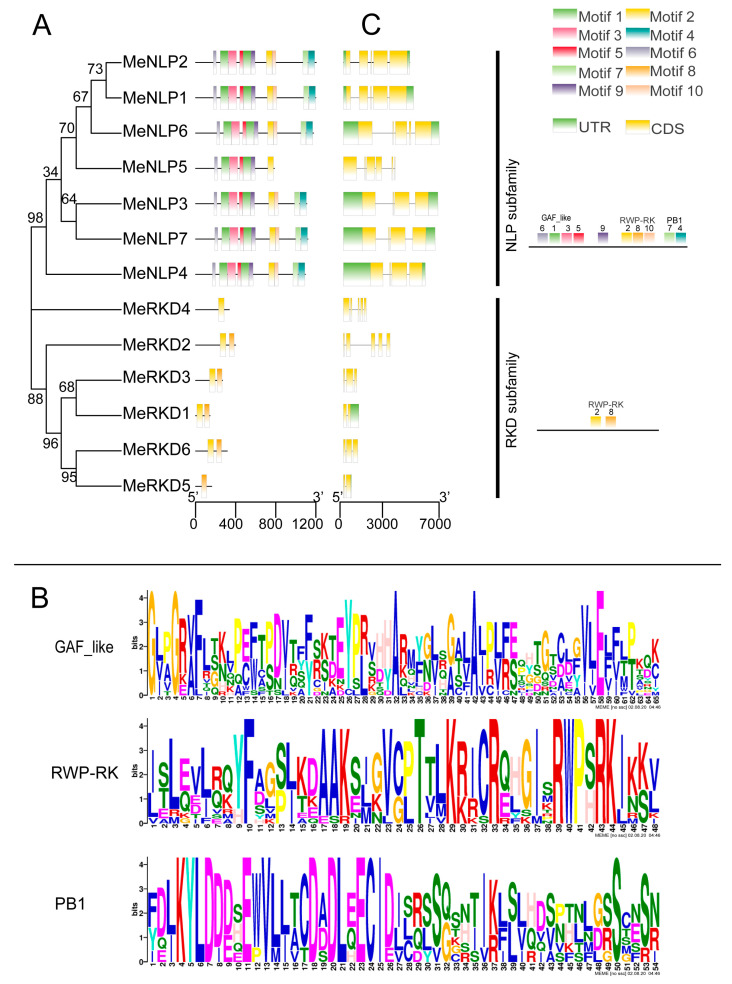
Motif and structure analysis of cassava RWP-RK family members. Note: (**A**), Protein motif analysis of cassava RWP-RK members. (**B**), Sequences of the three conserved motifs identified in this study. (**C**), Structure analysis of cassava *RWP-RK* genes. The green blocks represent the untranslated region (UTR), the yellow blocks represent exons, and the black lines represent introns.

**Figure 6 ijms-24-12925-f006:**
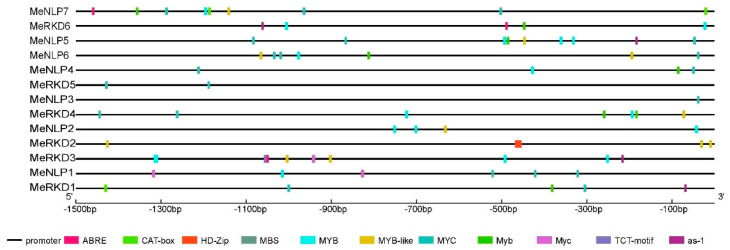
Prediction of cis-acting elements of *RWP*-*RK gene* promoter in cassava. Different cis-acting elements are represented in different colors. The same cis-acting elements of MYB, MYB−like, and Myb have different same because they derived from different plants, and MYC and Myc are also derived from different plants.

**Figure 7 ijms-24-12925-f007:**
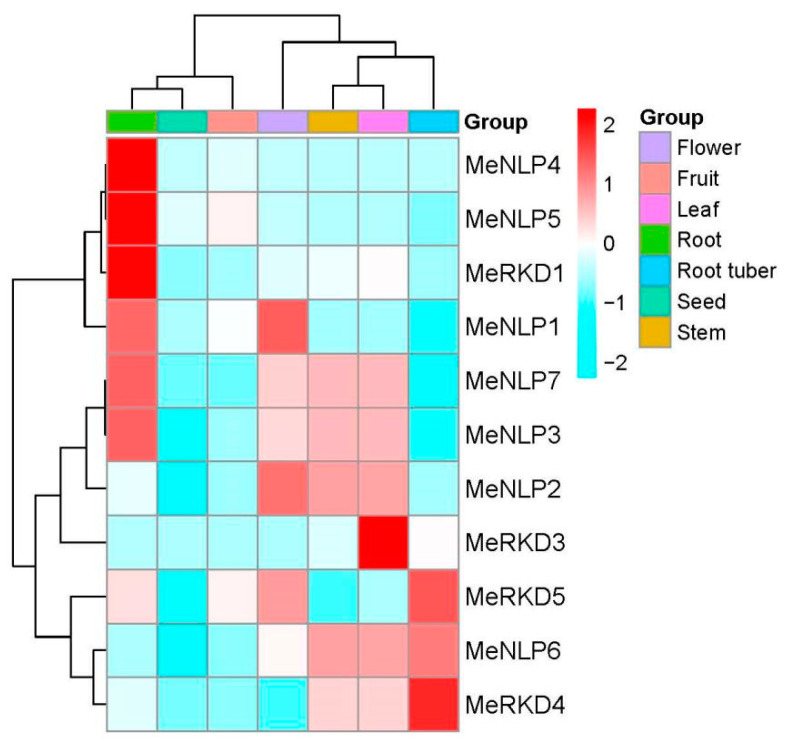
Tissue expression pattern of cassava *RWP-RK* genes in cassava. Log2 transformed FPKM value was used to create the heat map. The scale represents the relative signal intensity of FPKM values.

**Figure 8 ijms-24-12925-f008:**
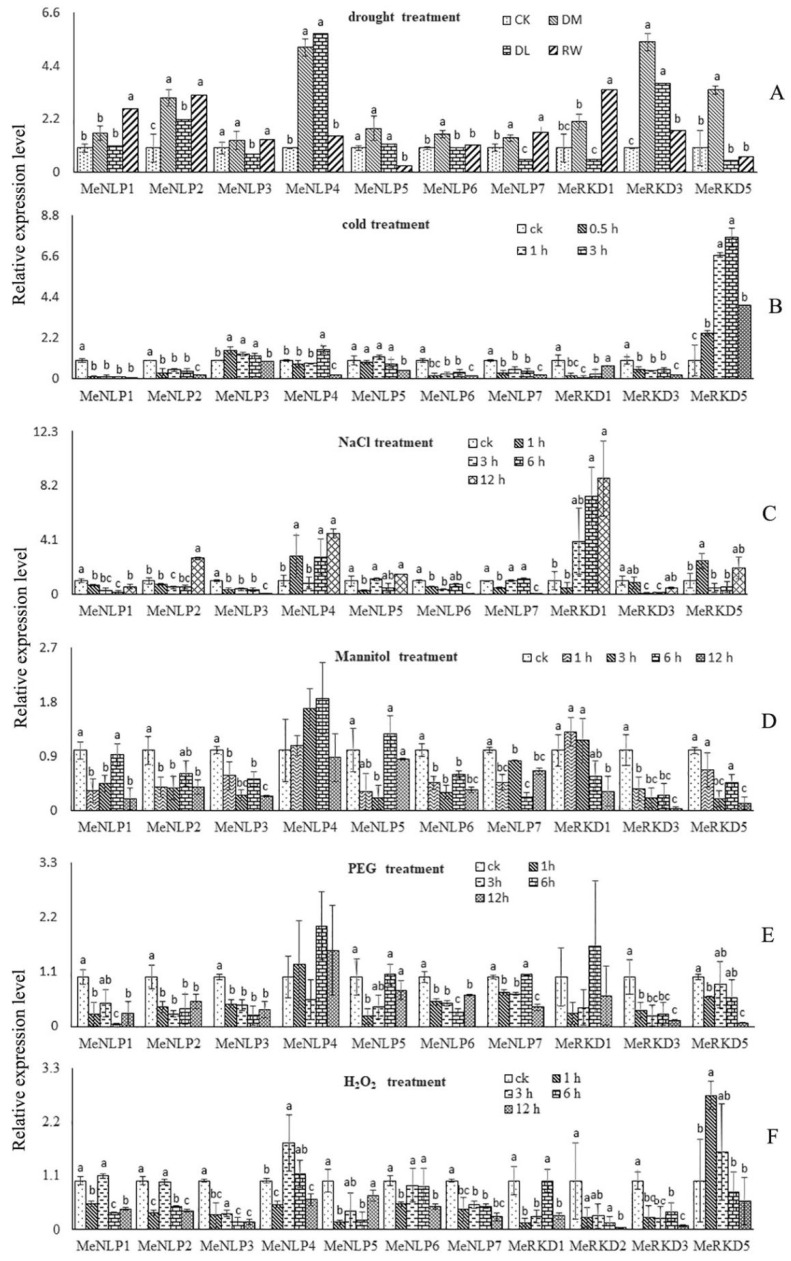
Expression analysis of *RWP-RK* genes in the leaves of cassava under different stresses. (**A**): drought treatment. DM, drought in mid; DL, drought in later stage; RW, re-watering. (**B**): cold treatment. (**C**): salt treatment. (**D**): Mannitol treatment. (**E**): PEG treatment. (**F**): H_2_O_2_ treatment. The error bars represent the standard error of the means of three independent replicates. Values denoted by the same letter did not differ significantly at *p* < 0.05 according to one-way ANOVA.

**Figure 9 ijms-24-12925-f009:**
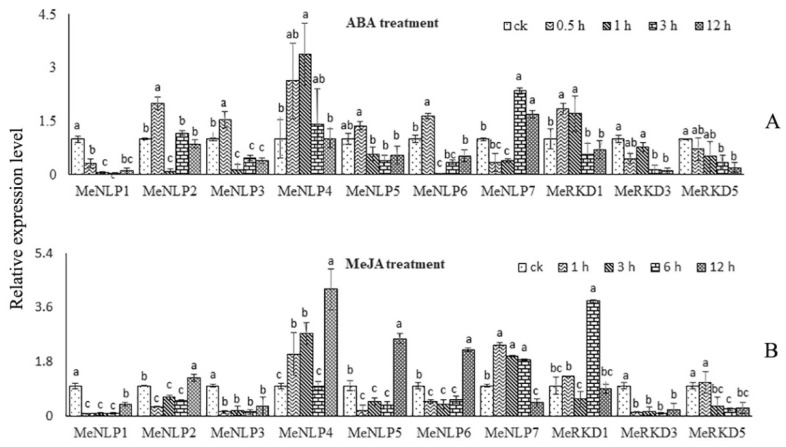
Expression analysis of *RWP-RK* genes in the leaves of cassava under different hormones. (**A**): ABA treatment. (**B**): JA treatment. The error bars represent the standard error of the means of three independent replicates. Values denoted by the same letter did not differ significantly at *p* < 0.05 according to one-way ANOVA.

**Figure 10 ijms-24-12925-f010:**
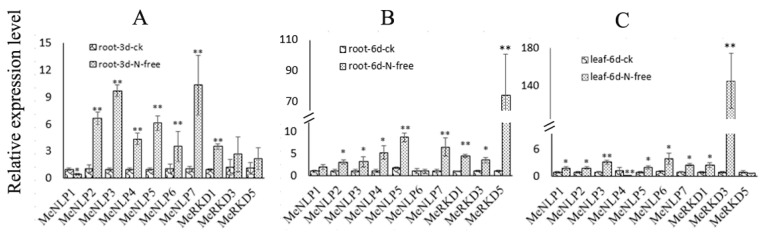
Expression profiles of different *RWP-RK* genes in the leaves and root of cassava under nitrogen starvation. (**A**): The third day of treatment in root. (**B**): I The sixth day of treatment in root. (**C**): The sixth day of treatment in leaf. The error bars represent the standard error of the means of three independent replicates. * and ** represent different at *p* < 0.05 and *p* < 0.01 according to Duncan’s multiple range tests.

**Figure 11 ijms-24-12925-f011:**
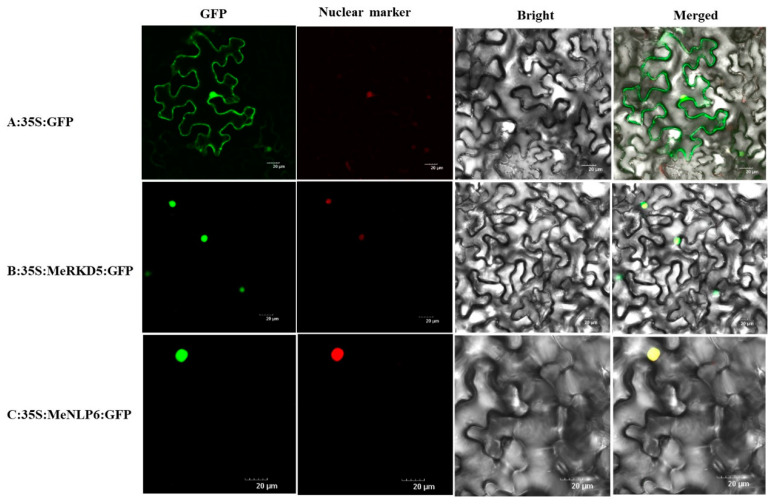
Fluorescent microscopic images of MeRKD5 and MeNLP6 in tobacco leaves. (**A**): 35S: green fluorescent protein (GFP). (**B**): 35S:MeNLP6: GFP. (**C**): 35S:MeRKD5: GFP.

**Table 1 ijms-24-12925-t001:** Screening of proteins interacting with NLP6/RKD5 by yeast two-hybrid system.

Protein Name	Identify	Gene Annotation	Amino Acid
MeNLP6	Manes.03G056700	PPR repeat family (PPR_2)	1498
MeRKD5	Manes.02G030100	Agamous-like mads-box protein AGL14	216
	Manes.01G174400	Metallothionein (Metallothio_2)	79
	Manes.08G023700	Zine finger FYVE domain containing protein	343
	Manes.05G125200	Glyceraldehyde-3-phosphate dehydrogenase	453
	Manes.15G059000	E3 Ubiquitin-protein ligase HUWE1	3649

## Data Availability

All data are available on reasonable request to the corresponding authors.

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
