# Peer review of "Genome-Wide Survey of the *RWP-RK* Gene Family in Cassava (*Manihot esculenta* Crantz) and Functional Analysis"

_ijms, 2023, doi:10.3390/ijms241612925_

Round 1

Reviewer 1 Report

Main comments:

The authors of the manuscript took up an interesting topic of genetic studies of the important plant cassava. The research includes the reaction to abiotic stresses, the use of selected plant growth inhibitors and nitrogen deficit in the substrate. The possibilities of using such broad, detailed analyses are very large, they can be used for further plant breeding and improvement of plant tolerance to abiotic stresses. Apart from a few shortcomings, the manuscript is well written, with detailed descriptions and figures. After correction and supplementation, it can be published.

Detailed comments and suggestions:

Line 24: Should be abiotic stresses (drought, cold, osmotic, salt and oxidative), or abiotic stressors (stress factors) such as water deficit, cold temperature, mannitol, polyethylene glycol, NaCl, and H2O2.

Line 470: What pots were used, their size and what substrate was used?

Line 473: Under what conditions did the plants grow, soil humidity, temperature and lighting?

Line 476: should be: months

Line 483: What type of PEG was used? (molecular weight, 8000?)

Line 535: Should be: 6 abiotic stressors (stress factors). Because PEG and mannitol cause osmotic stress.

I did not notice a significant lexical and syntax errors in the English language in the manuscript. The vocabulary and syntax used appear to be correct. 

Reviewer 2 Report

Dear Authors,

Reviewer comments ijms-2554712

The manuscript entitled „Genome-wide survey of the RWP-RK gene family in cassava (Manihot sculenta Crantz) and functional analysis“ represents a useful comprehensive study on RWP-RK transcription factor family in Manihot esculenta encompassing phylogenetic analysis, synteny analysis, promoter and sequence motif analyses, expression analysis under nitrogen starvation, fluorescent microscope image analysis, and protein interaction analysis of NLP6/RKD5 using Y2H approach.

I can conclude that the present manuscript represents a valueable overview on manihot RWP-RK gene family including its structure, expression patterns, subcellular localization and interactome analyses.

I can recommend the present manuscript for publication in International Journal of Molecular Sciences (IJMS). I have only a few comments on the present manuscript which have to be addressed by the authors:

In Figure 2 providing the phylogenetic analysis of RWP-RK proteins in Manihot esculenta, Arabidopsis thaliana and Oryza sativa as model dicotyledonous and monocotyledonous species, respectively, appropriate explanation of the numbers at nodes has to be added in the figure legend, i.e., probably „the numbers at nodes represent bootstrap values per 1000 replicates determined by Neighbor-Joining (N-J) method.“

In Materials and methods, line 422, for RWP-RK protein sequence search using NCBI database, the date of access, taxonomy (Manihot esculenta?), and the number of protein sequences employed for the search have to be specified.

In Materials and methods, for each kind of software named, i.e., „MapChart siftware“ (l. 434), „MCScanX“ (l. 446), „CIRCOS“ (l. 447), appropriate reference, i.e., web address and date of access have to be given.

Formal comments on the text:

Abstract, line 32: Use rather the word „basis“ instead of „foundation“ in the statement „These results provided a solid basis to understand abiotic stress responses…“

Introduction, line 52: Add a space between the word „differentiation“ and the following reference „[5].“

Discussion, line 349: Modify the word form „belonged“ to „belonging“ in the statement „….that the two plants belonging to dicotyledonous plants…“

Line 370: Replace the word „of“ with „between“ in the statememnt „Thus, some orthologous pairs formed after the divergence between dicotyledonous and monocotylednonous plants.“

Line 372: Modify the word form „In generally“ to „In general,…“

Line 377: Modify the word form in the statement „…were the highest than those in the other parts…“ to „….were higher than those in the other parts…“

Line 420: Modify the section heading „4. Methods“ to „4. Materials and methods“.

Line 447: Correct the term „the syntenic map“ (not „the syntonic map“).

Line 471: Add „plant growth stage“ in the dstatement „…in later (DL) plant growth stage“.

Line 479: Correct the word form in „Treated leaves“ (not „Treat leaves“).

Line 518: Add a comma before and after the word „respectively“, i.e., „Two cDNA fragments encoding MeNLP6 and MeRKD5, respectively, were amplified and inserted into plasmid pGBKT7…“

Line 521: Modify the statement as follows: „…and the cells were grown on SD-Trp-Leu medium for 3 days at 30°C…“

Line 542: Replace the word „in“ with „for“ in the statement „…these findings may be beneficial for studying RWP-RK gene function in cassava.“

Final recommendation: Accept after a minor revision.

Dear Authors,

Reviewer comments ijms-2554712

The manuscript entitled „Genome-wide survey of the RWP-RK gene family in cassava (Manihot sculenta Crantz) and functional analysis“ represents a useful comprehensive study on RWP-RK transcription factor family in Manihot esculenta encompassing phylogenetic analysis, synteny analysis, promoter and sequence motif analyses, expression analysis under nitrogen starvation, fluorescent microscope image analysis, and protein interaction analysis of NLP6/RKD5 using Y2H approach.

I can conclude that the present manuscript represents a valueable overview on manihot RWP-RK gene family including its structure, expression patterns, subcellular localization and interactome analyses.

I can recommend the present manuscript for publication in International Journal of Molecular Sciences (IJMS). I have only a few comments on the present manuscript which have to be addressed by the authors:

In Figure 2 providing the phylogenetic analysis of RWP-RK proteins in Manihot esculenta, Arabidopsis thaliana and Oryza sativa as model dicotyledonous and monocotyledonous species, respectively, appropriate explanation of the numbers at nodes has to be added in the figure legend, i.e., probably „the numbers at nodes represent bootstrap values per 1000 replicates determined by Neighbor-Joining (N-J) method.“

In Materials and methods, line 422, for RWP-RK protein sequence search using NCBI database, the date of access, taxonomy (Manihot esculenta?), and the number of protein sequences employed for the search have to be specified.

In Materials and methods, for each kind of software named, i.e., „MapChart siftware“ (l. 434), „MCScanX“ (l. 446), „CIRCOS“ (l. 447), appropriate reference, i.e., web address and date of access have to be given.

Formal comments on the text:

Abstract, line 32: Use rather the word „basis“ instead of „foundation“ in the statement „These results provided a solid basis to understand abiotic stress responses…“

Introduction, line 52: Add a space between the word „differentiation“ and the following reference „[5].“

Discussion, line 349: Modify the word form „belonged“ to „belonging“ in the statement „….that the two plants belonging to dicotyledonous plants…“

Line 370: Replace the word „of“ with „between“ in the statememnt „Thus, some orthologous pairs formed after the divergence between dicotyledonous and monocotylednonous plants.“

Line 372: Modify the word form „In generally“ to „In general,…“

Line 377: Modify the word form in the statement „…were the highest than those in the other parts…“ to „….were higher than those in the other parts…“

Line 420: Modify the section heading „4. Methods“ to „4. Materials and methods“.

Line 447: Correct the term „the syntenic map“ (not „the syntonic map“).

Line 471: Add „plant growth stage“ in the dstatement „…in later (DL) plant growth stage“.

Line 479: Correct the word form in „Treated leaves“ (not „Treat leaves“).

Line 518: Add a comma before and after the word „respectively“, i.e., „Two cDNA fragments encoding MeNLP6 and MeRKD5, respectively, were amplified and inserted into plasmid pGBKT7…“

Line 521: Modify the statement as follows: „…and the cells were grown on SD-Trp-Leu medium for 3 days at 30°C…“

Line 542: Replace the word „in“ with „for“ in the statement „…these findings may be beneficial for studying RWP-RK gene function in cassava.“

Final recommendation: Accept after a minor revision.
